



# Dynamic characteristics of snowfall particles in atmospheric turbulent boundary layer and its effect on dust wet deposition

Jie Zhang[1,2], Wanzhi Li[1,2], Ning Huang[1,2], Binbin Pei[1,2]

[1]College of Civil Engineering and Mechanics, Lanzhou University, Lanzhou, 730000, China
[2]Key Laboratory of Mechanics on Disaster and Environment in Western China, The Ministry of Education of China, Lanzhou, 730000, China

*Correspondence to*: Jie Zhang (zhang-j@lzu.edu.cn)

**Abstract.** Wet deposition by snowfall refers to the scavenging of atmospheric dust by snow particles. Existing models only consider vertical scavenging in quiescent atmosphere, neglecting the complex vertical and horizontal motion of snowfall particles induced by turbulence in the actual atmosphere boundary layer, affecting the accurate estimation of wet deposition flux. However, precise quantitative analysis of dust collection mechanism during snow particle setting remains lacking under turbulence. Therefore, we employ the Euler-Lagrange numerical method to simulate and analyze snow particles dynamic characteristics and dust collection in turbulent boundary layers. It is shown that increasing friction velocity ($u_*$) alters the dominant factors controlling the relative motion between snow particles and air. The transition occurs at a critical dimensionless parameter $\alpha_d = V_t/\kappa u_* = 0.2$ ($V_t$ is the terminal settling velocity of snow particles, and $\kappa=0.4$ is the von Kármán constant). When $\alpha_d > 0.2$, the vertical relative motion dominates, and its dominance strengthens with increasing $\alpha_d$; when $\alpha_d < 0.2$, the horizontal relative motion becomes predominant. This change in dynamic characteristics significantly enhances total dust collection capacity and shifts the dominant collection mechanism from vertical to horizontal: for $\alpha_d \geq 1$, vertical collection accounts for over 75% of the total, while under horizontal dominance, its contribution exceed 50%. The results show that neglecting horizontal collection underestimates wet deposition flux. Thus, we establish a quantitative wet deposition model, providing a theoretical basis for snowfall particle collection mechanisms under turbulent, with significant applications for predicting atmospheric dust wet deposition and artificial dust removal.

## 1 Introduction

Atmospheric dust particles influence climate through mechanisms such as absorption or reflection of radiation and modification of cloud properties (Fuzzi et al., 2015; Rosenfeld et al., 2014), while also contributing to respiratory and cardiovascular diseases (Dockery and Stone, 2007; Tang et al., 2017). Variations in dust concentration are primarily governed by the removal rate via atmospheric wet deposition (Seinfeld, 2016), which occurs through in-cloud and below-cloud scavenging (also referred to as the "washout" process). The contribution of below-cloud scavenging to total wet deposition range from 50% to 60% (Ge et al., 2021) and is considered a primary sink for source aerosols (Textor et al., 2006). The



efficiency of below-cloud scavenging is closely associated with the dynamic behaviours of precipitation particles (rain drops or snowfall particles): snowfall particles capture airborne dust through dynamic processes such as Brownian diffusion and inertial impaction, leading to episodic, high-flux precipitation and dust deposition events (Henzing and Olivie, 2006).

The dynamic behaviour of precipitation particles in the turbulent boundary layer forms the basis for studying dust wet deposition, with their motion governed by the interplay between particle properties and surrounding turbulence. Dorgan and Loth (2004) demonstrated that in a horizontal turbulent boundary layer, as particle size increases, the dominant mechanism of particle motion shifts from turbulent diffusion to gravitational settling. This transition fundamentally arises from a reversal in the relative dominance of turbulent fluctuation forces and gravity acting on the particles. Studies have found that for small heavy particles fall through turbulence, their mean settling velocity can be significantly altered compared to quiescent fluid

conditions (Wang and Maxey 1993;Balachandar and Eaton 2010). The fall can be hindered, e.g. if weakly inertial particles are trapped in vortices (Tooby et al., 1977), or if fast-falling particles are slowed down by nonlinear drag (Mei et al., 1991), or loiter in upward regions of the flow (Good et al., 2012). Field observations of snowfall events with significant variations in turbulence intensity by Li et al. (2021) revealed that turbulence is a critical factor influencing the falling velocity and spatial distribution of snowfall particles. Consequently, the regulatory effect of turbulence on the snowfall particle settling process

significantly influences the efficiency of dust particles scavenging.

The scavenging coefficient $\Lambda$, representing the rate of change in aerosol mass concentration due to below-cloud scavenging. (Seinfeld and Pandis, 2016):

$$\frac{dN(d_p)}{dt} = -\Lambda(d_p)N(d_p), \tag{1}$$

$$\Lambda(d_p) = \int_0^\infty A(V_D - v_d)E(d_p, D_p)N(D_p)dD_p, \tag{2}$$

Where $N(d_p)$ is the mass concentration of aerosols with particle diameter $d_p$, and $N(D_p)dD_p$ is the number of precipitation particles with diameter between $D_p$ and $D_p+dD_p$ in a unit volume of air (m$^{-3}$). $V_D$ and $v_d$ are the terminal velocities (m s$^{-1}$) of precipitation particles and dust particles, respectively, $E(d_p, D_p)$ is the collection efficiency (dimensionless) between dust particles of size $d_p$ and a precipitation particle of size $D_p$, and $A$ is the effective cross-sectional area of precipitation particles projected normal to the fall direction (m$^2$). The settling velocity and collection efficiency of precipitation particles, among

other key parameters, are closely related to their dynamic characteristics during fall. Previous experimental and theoretical results have shown that $E(d_p, D_p)$ is the results of net action of various forces influencing the relative motion of aerosols and precipitation particles. (Langmuir, 1948) first proposed the theoretical framework for "collision efficiency" through a dynamic model. Subsequently, Greenfield (1957) elucidated the effects of Brownian diffusion, interception, and inertial collision as three key mechanisms influencing $E(d_p, D_p)$. Building on this, Slinn and Hales (1971) integrated theoretical and

observational data, developing the widely applied Slinn formula. However, classical models assume a quiescent fluid condition, neglecting the effects of turbulence on particle transport. As research on multiphase flow dynamics has advanced, the





regulatory role of turbulence in the motion of precipitation particles has been recognized to further influence the collection efficiency $E(d_p, D_p)$ of dust aerosols. In recent years, studies on the mechanisms affecting $E(d_p, D_p)$ have begun to incorporate the roles of physical processes such as atmospheric turbulence and the wake effects of raindrops. Hua et al. (2017)
found that, in the context of atmospheric turbulence, the randomness of particle trajectories increases due to pulsating airflow, thereby enhancing the collection efficiency $E(d_p, D_p)$ of stationary raindrops for aerosol particles, providing a new definition of aerosols capture by raindrop under the turbulent effect.

Compared to raindrops, research on the aerosol scavenging efficiency of snowfall particles remains relatively scarce. The wide variety of snow-particle shapes, sizes, and density, which results in significantly different terminal falling velocity, cross-
sectional area, and surrounding flow field structure, leading to greater uncertainty in the scavenging coefficient (Zhang et al., 2013). Currently, relevant research can be divided into two type. The first type uses field-collected snow sample data to establish relationships between precipitation, snowfall particle size distribution, and ground-level aerosol mass concentration, thereby evaluating the scavenging efficiency of snowfall on aerosol. The second type focuses on the collision scavenging mechanisms between a single snowflakes (or snow crystals) falling at terminal velocity $V_D$ and aerosol particles, investigated
through laboratory simulations or direct observations (Pruppacher et al., 1998). Studies have found that the terminal velocity of snowfall particles is a critical dynamical parameter determining scavenging efficiency. To quantitatively describe the evolution of this parameter, early research on snowfall particle terminal velocity established an empirical power-law formula based on the maximum dimension of snowfall particles through experimental measurements (Langleben, 1954). Subsequent studies progressively refined theoretical models (Mitchell and Heymsfield, 2005; Pruppacher and Klett, 1997), with the
formula proposed by Mitchell and Heymsfield becoming the benchmark for current research due to its general applicability. The distinct physical reasons for the influence of $V_D$ and the cross-sectional area ($A$) on the scavenging coefficient are straightforward: a higher fall velocity or a larger cross-sectional area of the collector (raindrop, snow, or ice) results in a faster collection process.

The existing scavenging coefficient expression is based on the assumption of a quiescent atmospheric environment and only
considers the relative motion between snowfall particles and dust in the vertical direction. However, due to a 1–3 orders of magnitude difference in their particle sizes, their aerodynamic behaviours exhibit significant disparities: dust particles follow turbulent eddy motions, whereas larger snowfall particles, with greater inertia driving gravitational settling, exhibit limited responsiveness to turbulence. In the atmospheric turbulent boundary layer, this inertial disparity induces significant horizontal relative motion between particles, thereby influencing the scavenging efficiency of snowfall particles for dust. To address the
limitations of traditional wet deposition models under quiescent condition in characterizing turbulence effects, this study employs numerical methods to dynamically reconstruct realistic turbulent fields, accurately simulating the motion trajectories of snowfall particles in turbulent environments. Based on Eulerian-Lagrangian coupled framework, we employ a gas-solid two-phase flow numerical simulations system that the OpenFOAM computational fluid dynamics platform with the Lagrangian particle tracking method. The system is used to simulate and analyse the motion behaviours of snowfall particles within the
atmospheric turbulent boundary layer and their relationship with turbulence characteristics. Our work reveals the influence



mechanisms of dust collection by snowfall particles and ultimately establishes a quantitative mathematical model for the snowfall scavenging process.

## 2.Numerical methods and validation

### 2.1 Flow field equations

In this study, atmospheric temperature variations are neglected, and the atmospheric boundary layer is assumed to be under neutral conditions. The three-dimensional incompressible Navier-Stokes (N-S) equations are employed to solve the turbulent flow field in the boundary layer, while the influence of energy exchange due to solar radiation on flow behaviour is disregarded. The mass conservation equation and momentum conservation equations implemented in OpenFOAM are expressed as:

$$\frac{\partial u_i}{\partial x_i} = 0, \tag{3}$$

$$\frac{\partial u_i}{\partial t} + u_j \frac{\partial u_i}{\partial x_j} = -\frac{1}{\rho}\frac{\partial p}{\partial x_i} + \nu \frac{\partial^2 u_i}{\partial x_j \partial x_j}, \tag{4}$$

where $u_i$ is the vector velocity of the fluid, while $P$, $\rho$ and $\nu$ represent pressure, air density, and kinematic viscosity, respectively.

 The hybrid RANS/LES strategy has been successfully applied to various complex flow scenarios, including the detailed resolution of high-Reynolds-number atmospheric boundary layer flows (Haupt et al., 2011), prediction of wind field characteristics around buildings (Liu and Niu, 2016), and investigation of near-field pollutant dispersion mechanisms (Lateb

et al., 2014), with its reliability substantiated through extensive validation cases. This hybrid strategy employs RANS statistical modelling for small-scale eddies near the wall:

$$\frac{\partial \overline{u_i}}{\partial x_i} = 0, \tag{5}$$

$$\frac{\partial \overline{u_i}}{\partial t} + \frac{\partial \overline{u_i}\,\overline{u_j}}{\partial x_j} = -\frac{1}{\rho}\frac{\partial \overline{p}}{\partial x_i} + \nu \frac{\partial^2 \overline{u_i}}{\partial x_j \partial x_j} - \frac{\partial \tau_{ij}^{RANS}}{\partial x_j}, \tag{6}$$

here, the overbar '—' denotes the time-averaged quantity, and $\tau_{ij}^{RANS}$ represents the Reynolds stress. In regions far from the

wall, the model transitions to LES for resolving large-scale turbulence:

$$\frac{\partial \tilde{u}_i}{\partial x_i} = 0, \tag{7}$$

$$\frac{\partial \tilde{u}_i}{\partial x_i} + \frac{\partial \tilde{u}_i \tilde{u}_j}{\partial x_j} = -\frac{1}{\rho}\frac{\partial \tilde{p}}{\partial x_i} + \nu \frac{\partial^2 \tilde{u}_i}{\partial x_j \partial x_j} - \frac{\partial \tau_{ij}^{LES}}{\partial x_j}, \tag{8}$$



where the superscript "~" denotes the resolvable-scale component of the physical quantity processed by the spatial filtering function. $\tau_{ij}$ represents the energy transfer between the filtered-out small-scale turbulence and the resolvable-scale turbulence, known as the subgrid-scale stress.

Due to the significant turbulence characteristics in the atmospheric boundary layer, accurate resolution of flow structures near the wall is required. We employ the Delayed Detached-Eddy Simulation (DDES) method based on the S-A model (Spalart et al., 2006). This method divides the simulation domain through the hybrid length scale $l_{DDES}$ and establishes an adaptive turbulence resolution framework:

$$l_{DDES} = d - f_d \max\{0, d - C_{DES}\Delta\}, \tag{9}$$

where $d$ represents the Euclidean distance from a flow field grid point to the nearest solid wall, $\Delta = max(\Delta x, \ \Delta y, \ \Delta z)$ is the filter width defined as the local maximum grid spacing in the three direction, and the coefficient $C_{DES}$=0.65. Spalart et al. (2006) introduced the shielding function $f_d$, improving the $l_{DDES}$ length scale based on the DES method (Spalart, 2000). DDES addresses the critical issue of premature transition from RANS to LES in traditional hybrid methods.

## 2.2 Snowfall particle motion equations

We employ the Lagrangian particle tracking method to solve the motion of snowfall particles (Huang et al., 2024; Li et al., 2016, 2018). Snowfall particles are treated as a dilute phase (volume fraction $\varphi_V < 10^{-6}$), considering only the effect of the fluid on the particles. As the particle diameter $D_p$ is smaller than the Kolmogorov length scale $\eta$, particle collisions and rotational motion are neglected (Balachandar and Eaton, 2010b). The shape of snowfall particles is simplified as spherical, with their motion governed by the following equation:

$$m_p \frac{d\bar{u}_p}{dt} = \frac{1}{6}\pi(\rho_p - \rho)D_p{}^3 g + \frac{1}{2}C_{Dp}A_p\rho|\bar{u}_f - \bar{u}_p|(\bar{u}_f - \bar{u}_p), \tag{10}$$

where $\vec{u}_p$ represents the velocity of snowfall particles; the snowfall particle density is $\rho_p$=340 kg m$^{-3}$; $g$=9.81 m s$^{-2}$ is the gravitational acceleration; and $D_p$ is the snowfall particle diameter. The combined force of gravity and buoyancy acting on the snowfall particle is $F_g + F_b = \frac{1}{6}\pi(\rho_p - \rho)D_p^3 g$. $C_{Dp}$ is the drag force coefficient for the snowfall particle; Re$_p$ is the particle Reynolds number, with their respective expressions given by:

$$C_{dp} = \begin{cases} \dfrac{24}{\text{Re}_p} + \dfrac{6}{1+\sqrt{\text{Re}_p}} + 0.4, & \text{Re}_p \leq 1000 \\ 0.424, & \text{Re}_p > 1000 \end{cases}, \tag{11}$$

$$\text{Re}_p = \frac{|\vec{u}_f - \vec{u}_p|D_p\rho}{\mu}. \tag{12}$$



The boundary condition for snowfall particles at the wall is set to trap, meaning that the particle trajectory calculation terminates when snowfall particles settle to the ground (wall). All other boundary conditions are set to escape.

**2.3 Case setup**

We conduct a systematic numerical simulation study on the motion of snowfall particles in the atmospheric boundary layer. The snowfall particle diameter range was set based on the average particle size reported by Li et al. (2021). By specifying different initial inlet velocities (1, 5, 12, 15, and 20 m s⁻¹), corresponding friction velocities ($u_*$=0.06, 0.31, 0.75, 0.93 and 1.18 m s⁻¹) were obtained, with a focus on investigating the motion characteristics of snowfall particles under different $u_*$ conditions. Considering that the scavenging coefficient is sensitive to particles dynamic processes within only a few hundred meters above the ground (Schumann, 1989), 2000 snowfall particles were released at 0.5 s intervals from a height of 100 m within a fully developed turbulent field. A time step of 0.05 s (CFL < 1) was set to ensure computational stability. Specific operating conditions and particle parameters are presented in Table 1.

**Table 1: Snowfall particle $\mathbf{Re}_p$ under different $u_*$ conditions**

| $D_p$(μm) | $u_*$=0.06 m s⁻¹ | $u_*$=0.31 m s⁻¹ | $u_*$=0.75 m s⁻¹ | $u_*$=0.93 m s⁻¹ | $u_*$=1.18 m s⁻¹ |
|---|---|---|---|---|---|
| 500 | 32.55 | 32.66 | 33.18 | 33.57 | 34.21 |
| 400 | 20.14 | 20.25 | 20.78 | 21.14 | 21.68 |
| 350 | 14.96 | 15.09 | 15.62 | 15.97 | 13.46 |
| 300 | 10.54 | 10.68 | 11.17 | 11.83 | 11.91 |
| 200 | 3.97 | 4.12 | 4.62 | 4.91 | 5.21 |
| 150 | 1.90 | 2.05 | 2.55 | 2.63 | 3.04 |
| 120 | 1.08 | 1.23 | 1.73 | 1.86 | 2.13 |
| 100 | 0.63 | 0.80 | 1.29 | 1.47 | 1.65 |
| 80 | 0.34 | 0.52 | 0.91 | 1.11 | 1.29 |
| 50 | 0.10 | 0.24 | 0.54 | 0.65 | 0.79 |

**2.4 Turbulent inflow validation**

To accurately simulate the turbulent wind field in the atmospheric boundary layer, it is necessary to establish inlet boundary conditions that conform to realistic turbulence characteristics. We construct a numerical wind tunnel based on the experimental model scale of Ishihara et al. (1999) (wind tunnel dimensions: $9 \times 0.6 \times 0.7$ m³, with $x$, $y$ and $z$ representing the streamwise, vertical, and spanwise directions, respectively, as shown in Fig. 1), where $h$=0.04 m. A precursor simulation strategy based on the recycling method (Lund et al., 1998; Nozawa and Tamura, 2002; Vasaturo et al., 2018) is adopted, with a recycle station placed at the central streamwise cross-section of the computational domain (Fig. 1). Velocity data from this cross-section are collected in real-time and reintroduced at the inlet as the turbulent velocity boundary (inflow). The outlet is set as a free outflow (Outflow), the top and side boundaries are set as symmetry planes (symmetry), and the bottom is a no-slip wall (No-slip wall) (Zhou et al., 2022). A wall function (Wang and Moin, 2002) is employed to accurately capture the turbulent evolution near the rough wall. To maintain turbulence self-sustainability, a uniform grid resolution is used in the streamwise direction of the computational domain, with local refinement near the wall, resulting in a total of 2.65 million grid cells (as shown in Fig. 2).



The wind profile generated by the recycling method is described by the logarithmic law:

$$\bar{u}(z) = (\frac{u_*}{\kappa}) \ln(\frac{z}{z_0}), \tag{13}$$

where $\kappa$ is the von Kármán constant with a value of 0.41, and $z_0$ is the aerodynamic roughness length. The root mean square

170    of the streamwise fluctuating velocity is given by:

$$\sigma^2 = \sum_{i=1}^{N} (u_i - \bar{u})^2 / (N-1), \tag{14}$$

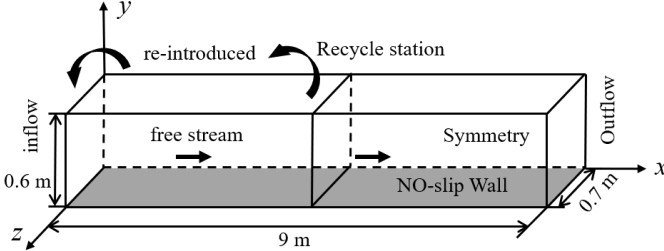

**Figure 1. Computational domain and boundary conditions.**

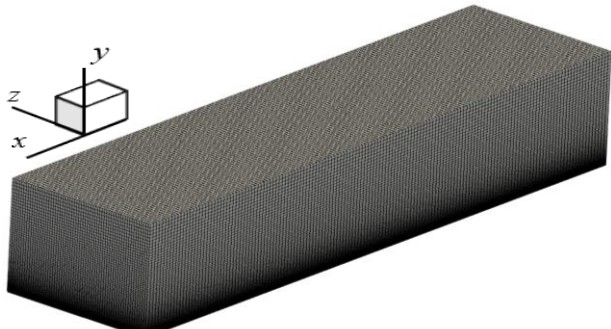

175    **Figure 2. Schematic of the overall grid division.**

The initial uniform inflow velocity is set to 5.2 m s$^{-1}$, with a time step of 0.001 s (satisfying CFL < 1). The turbulent field

reaches a fully developed state after 10 s, and thus, the computational results from the 10–20 s time interval are selected for

time-averaged statistical analysis. The fluctuating wind speed at the monitoring point exhibits typical random characteristics

(Fig. 3). The simulated profiles of mean wind velocity and turbulence fluctuation based on these results are shown in Fig. 4.

180    The simulation results of this study are compared with the numerical results of Zhou et al. (2022) and the experimental data of

Ishihara et al. (1999).

The mean wind velocity is in good agreement with the simulated values of Zhou et al. (2022) and the experimental data,

with a maximum relative error of approximately 4%. The simulated standard deviation of the streamwise fluctuating velocity



($\sigma_u/U_{ref}$) is generally consistent with literature values and experimental data in the region $y/h > 1.5$, with localized deviations
near the wall (the maximum relative error in turbulence fluctuations is approximately 15%). The power spectrum shows good
agreement with the simulation results of Zhou et al. (2022). Furthermore, the Reynolds number in this study (Re=3.12×10⁶) is
in exact agreement with the simulated value of Zhou et al. (2022), while the wind tunnel experimental value is 2.8×10⁶.
Therefore, we adopt the recycling method based on the DDES turbulence model, which can accurately reproduce the
characteristics of the atmospheric boundary layer turbulent wind field, validating the reliability and effectiveness of the method.

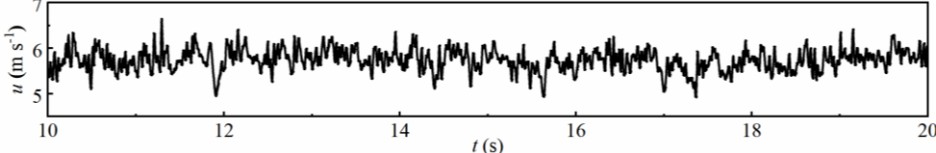

**Figure 3.** Time history of fluctuating wind velocity at the monitoring point ($x = 6$ m, $y = 0.6$ m).

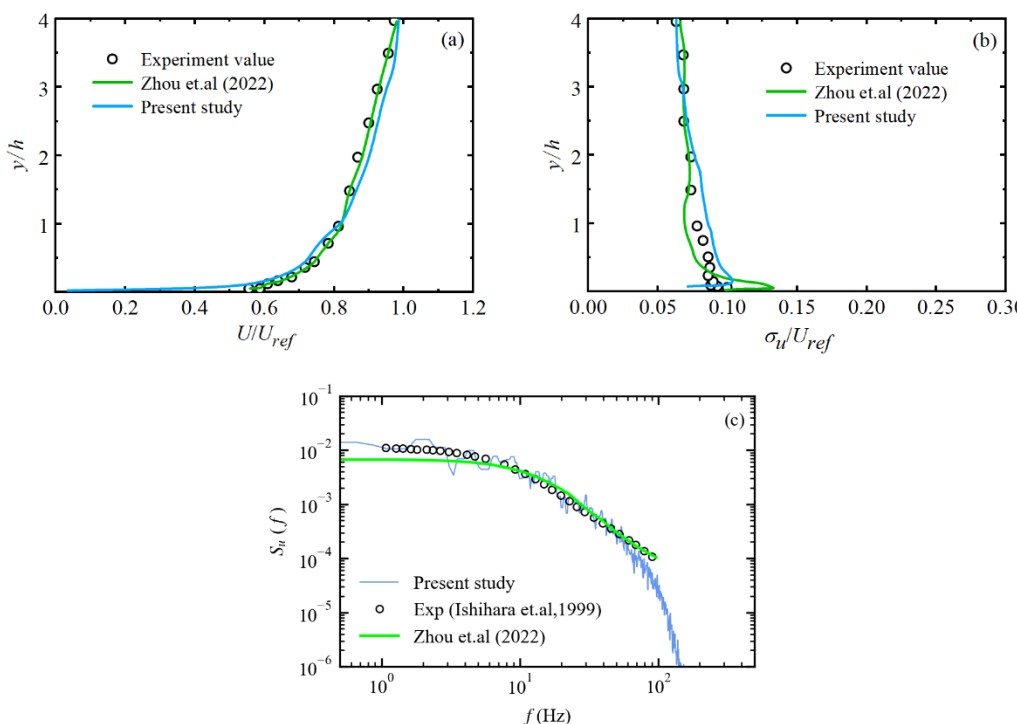

**Figure 4. Characteristics of the turbulent boundary layer for the empty wind field obtained through numerical simulation, (a) normalized mean wind velocity, (b) normalized turbulent fluctuation, (c) power spectrum (streamwise).**

## 2.5 Grid independence verification

To reproduce the realistic atmospheric boundary layer, a numerical model of the empty wind field is established with
dimensions: length ($x$) × height ($y$) × width ($z$) = 2000 × 400 × 600 m³. The boundary conditions are consistent with the



previously validated case. To verify the reliability and accuracy of the numerical results, a grid independence study was conducted (Roache, 1993). The Grid Convergence Index (GCI) is defined as:

$$GCI = F_s \frac{|\varepsilon|}{r^m - 1},$$ (15)

where $\varepsilon=(f_1-f_2)/f_1$ represents the relative error of grid convergence ($f_1$ and $f_2$ are the solutions from the fine grid and the previous coarse grid, respectively); the grid refinement ratio is $r_{h,k+l}=(N_{k+l}/N_k)^{1/3}$, with $N$ denoting the number of grid nodes; the safety factor $F_s$=1.25. We employ a second-order accurate scheme, thus m=2.

We employ five grid schemes with different resolutions. Taking the calculated total relative travel distance ($Sr$) of snowfall particles with an initial velocity of 12 m s$^{-1}$ in air as an example, the GCI results are presented in Table 2. Using the criteria of $\varepsilon \leq 1\%$ and GCI≤5% GCI, for grid numbers of 51.75 million, 68.48 million, and 79.17 million, the relative errors of $Sr$ for snowfall particles with $D_p$=500 μm are all less than 1%, with corresponding GCI values of 4.09% and 3.69%, respectively. Considering both computational efficiency and resource costs, a grid number of 51.75 million is ultimately selected for the simulations.

**Table 2. GCI Results for $Sr$ of snowfall particles ($D_p$=500 μm)**

| Grid Number ($\times 10^6$) | Grid Nodes ($\times 10^6$) | $r$ | $Sr_{500}$ | $\varepsilon_{500}$ | $GCI_{500}(\%)$ |
|---|---|---|---|---|---|
| 10.53 | 10.69 | 1.46 | 100.65 | 0.035 | 3.82 |
| 32.76 | 33.10 | 1.16 | 104.26 | -0.027 | 9.72 |
| 51.75 | 52.21 | 1.10 | 101.525 | 0.007 | 4.09 |
| 68.48 | 69.04 | 1.05 | 102.229 | 0.003 | 3.69 |
| 79.17 | 79.79 | — | 102.54 | — | — |

## 3. Numerical results

### 3.1 Statistical characteristics of the wind field

The wind field simulation results based on the aforementioned grid scheme (Fig. 5) demonstrate that the simulated atmospheric boundary layer turbulent wind field exhibits the irregularity and randomness characteristic of the real atmosphere. Figure. 6 shows the corresponding mean wind velocity and turbulence intensity profiles. Comparisons with the international standard ASCE7-III (Zhou and Kareem, 2002) and the Chinese standard (Load Code for Design of Building Structures, GB 50009-2012) reveal that: the simulated mean wind velocity profile fall between the value of the Chinese standard and the international standard.; the turbulence intensity profile lies between the Chinese standard and the upper limit of turbulent intensity in ASCE7-III, and are essentially consistent with the lower limit of turbulence intensity in ASCE7-III (ASCE7-III turbulence intensity values fluctuate within a ±20% range of the recommended baseline value (Kozmar, 2011)), with a





maximum relative error of 16% observed in the near-surface region. In summary, the simulation results effectively reproduce the wind field characteristics of the real atmospheric boundary layer.

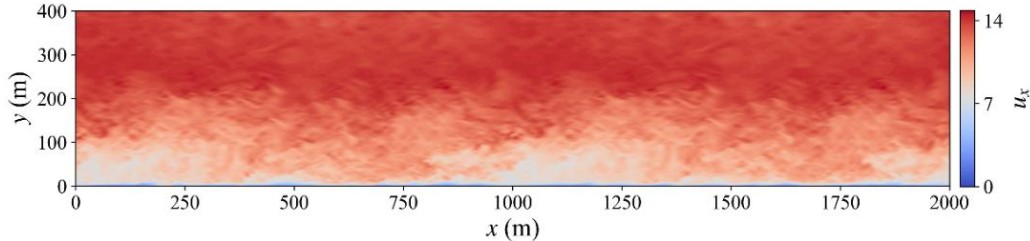

**Figure 5. Instantaneous streamwise velocity distribution in the x-y plane of the turbulent wind field.**

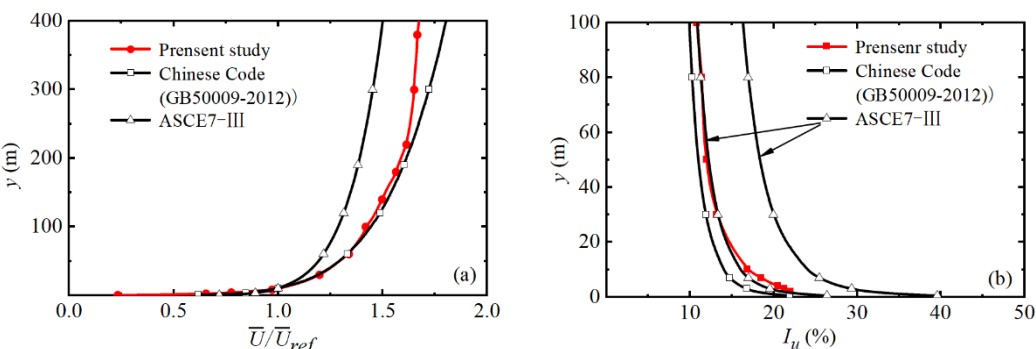


**Figure 6. Comparison of (a) the profile of the ratio of measured mean wind velocity to the mean wind velocity at the reference height of 10 m, and (b) the turbulence intensity profile with the Chinese standard and the international standard.**

### 3.2 Analysis of snowfall particle dynamic characteristics

Analysis of the motion trajectory data over a 1000 s duration indicates that larger-diameter snowfall particles, due to their greater mass and inertia, primarily exhibit stable vertical settling with a weak response to turbulent disturbances. In contrast, the motion of smaller-diameter snowfall particles is governed by both gravity and turbulent diffusion: gravity dominates vertical motion, while horizontal transport induced by turbulent eddies significantly broadens their settling range and increases uncertainty in settling positions (Fig. 7a). As the friction velocity increases, the motion trajectories of snowfall particles (particularly those with smaller diameters) are more strongly affected by turbulent disturbances, displaying more pronounced perturbations and distortions (Fig. 7b). We characterize the dynamic state of snowfall particles in the airflow by referencing the ratio $\alpha_d$ ($=V_t/w$), where $V_t$ is the terminal settling velocity of the snowfall particles and $w$ ($w=\kappa u_*$, with $\kappa=0.4$ as the von Kármán constant) is the vertical diffusion velocity of the fluid (Huang and Shi, 2017; Scott, 1995). This parameter reflects the competition between gravitational settling and airflow disturbances. The terminal settling velocity $V_t$ can be calculated using the following equation(Carrier, 1953):





$$V_t = -\frac{A}{D_p} + \sqrt{\left(\frac{A}{D_p}\right) + BD},$$


$$A = 6.203\nu_a,$$ (16)

$$B = \frac{5.516\rho_p}{8\rho_a}g,$$

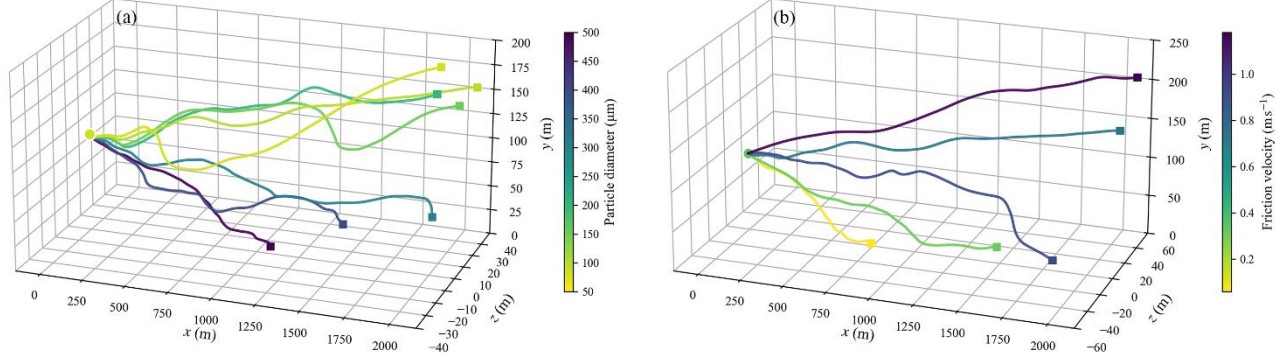

**Figure 7. Random motion trajectories of (a) snowfall particles with different diameters at $u_*$=0.75 m/s and (b) snowfall particles ($D_p$=500 μm) under different $u_*$ conditions.**

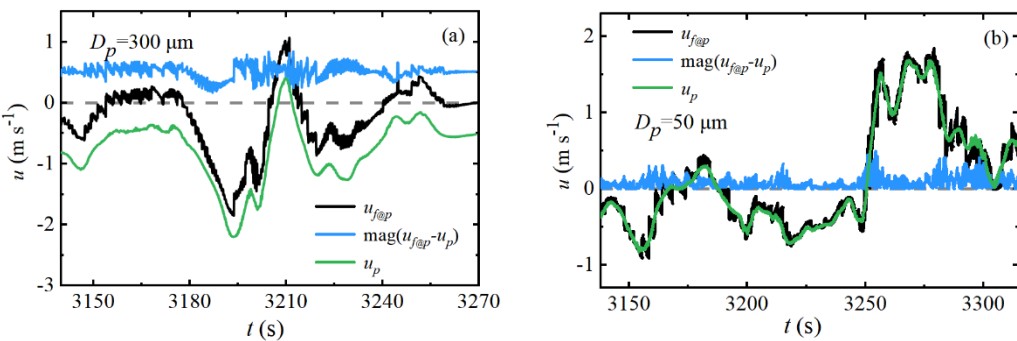

**Figure 8. Temporal variations of snowfall particle velocity, wind field velocity at the snowfall particle location, and snowfall particle relative velocity $Vr$ under $u_*$=0.75 m/s.**

Fig. 8 shows the temporal evolution curves of $u_{f@p}$ (where $u_{f@p}$ is the fluid velocity at the particle's location), $|u_{f@p}-u_p|$, and $u_p$ for snowfall particles of two different diameters during their falling process. The speed of the snowfall particle relative to the wind field at its location is defined as the relative velocity of the snowfall particle:

$$Vr = \left|u_{f@p} - u_p\right|,$$ (17)

The results indicate that the local flow fields experienced by snowfall particles of different diameters exhibit significant differences. Snowfall particles with $D_p$=300 μm demonstrate a stronger resistance to flow field disturbances, with higher



relative speed (*Vr*) and more independent motion. In contrast, smaller-diameter snowfall particles ($D_p$=50 μm) exhibit significantly lower *Vr*, making them more susceptible to turbulence and displaying greater flow-following behaviour. This

difference in *Vr* reflects the distinct dynamic effects of the flow field on snowfall particles of varying diameters. Further analysis reveals that the distribution of $Vr/V_t$ for snowfall particles follows a normal distribution, with the distribution shape remaining unaffected by changes in $u_*$ and $D_p$ (Fig. 9) . The parameters $u_*$ and $D_p$ primarily influence the characteristic quantities (mean and variance) of the normal distribution.

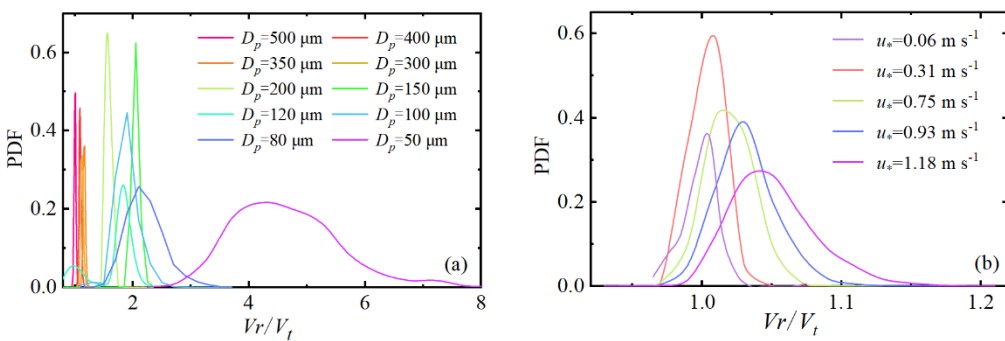

**Figure 9. Probability distribution of $Vr/V_t$ for (a) snowfall particles with Different diameter at $u_*$=0.75 m/s and (b) snowfall particles with $D_p$=500 μm under different friction velocities.**

To characterize the influence of different $u_*$ on the relative motion velocity of snowfall particles with varying diameter, Fig.10 and 11 show the mean and standard deviation of the relative velocity as functions of friction velocity for different particle diameters. As shown in Figure 10a, the dimensionless mean relative velocity ($Vr/V_t$ ) increases exponentially with

increasing friction velocity. Both parameters *a* and *b* decreased exponentially with increasing snowfall particle diameter (Fig. 10b). The variation of the mean relative velocity of snowfall particles with friction velocity and particle diameter can be expressed by Eq. (18):

$$Vr = V_t \cdot (0.97 + 19.85e^{-57.6D_p})e^{2.27e^{-11.02D_p} \cdot u_*}, \tag{18}$$

As shown in Fig. 11a, the standard deviation of the relative velocity fluctuations ( $Vr'$ ) increases linearly with increasing

friction velocity. The intercept of the linear function remains independent of snowfall particle diameter, while the parameter $a'$ exhibits negative exponential growth with increasing particle diameter. The variation of $Vr'$ with friction velocity and snowfall particle diameter can be expressed by Eq. (19):

$$Vr' = 0.003 + \left(0.17 - 0.05e^{-3.25D_p}\right)u_*, \tag{19}$$

$$Vr = Vr + Vr'. \tag{20}$$





The coefficient of determination ($R^2$) for the mean relative velocity exceeds 0.90, while that for the fluctuating standard deviation of relative velocity surpasses 0.99. Consequently, Eq. (20) should be used for predicting snowfall particle relative velocity.

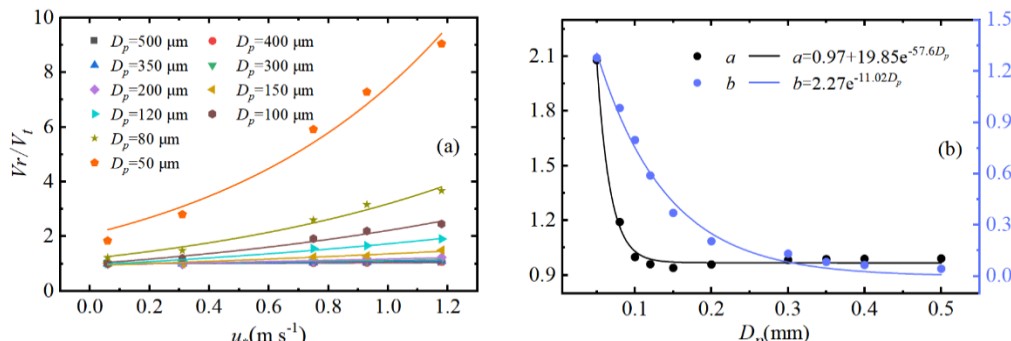

**Figure 10. (a) Dimensionless mean relative velocity of snowfall particles with different $D_p$ under various $u_*$ conditions, and (b) the**
**corresponding fitting parameters as a function of $D_p$ (symbols represent simulated values, and solid lines represent fitted values).**

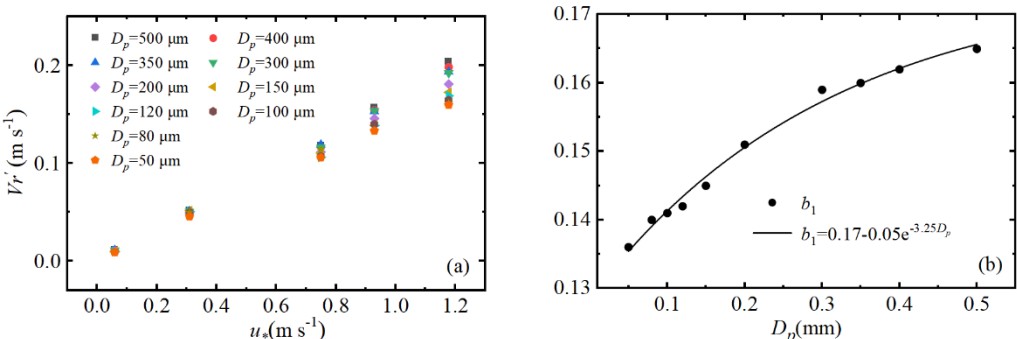

**Figure 11. (a) Standard deviation of the fluctuating relative velocity of snowfall particles with different $D_p$ as a function of $u_*$, and (b) the corresponding fitting parameters as a function of $D_p$.**

Limited by the difficulty of fully simulating the trajectories of small-diameter particles, we introduce the product of the
snowfall particle relative velocity $Vr$ (for which a quantitative characterization method has been established) and the particle suspension time to quantify the relative travel distance of particles during their airborne phase. The mean particle suspension time is calculated based on the deposition velocity ($V_d$) theoretical framework of the two-layer model proposed by Zhang and Shao (2014):

$$V_d = \left( r_g + \frac{r_s - r_g}{\exp(r_a / r_g)} \right)^{-1},$$    (21)

here, the gravitational resistance is denoted as $r_g = 1/V_t$, the aerodynamic resistance as $r_a = (B_1 \cdot Sc_T / \kappa u_*)\ln(z/z_0)$, and the surface collection resistance as $r_s = (w_{dm}(Sc^{-1} + 10^{-3/Tp+}) + V_{t,\delta})^{-1}$. Where $B_1$ is an empirical constant, $Sc$ and $Sc_T$ are the Schmidt number





and turbulent Schmidt number, respectively, $w_{dm}$ is the conductance for momentum, $T_p^+$ Is the dimensionless particle relaxation time, and $V_{t,\delta}$ is the terminal velocity of particles at the top of the laminar layer, typically approximated as $V_t$. Fig. 12a shows a comparison between the theoretical and simulated values of snowfall particle deposition velocity $V_d$. At $u_*=0.75$ m s$^{-1}$, the

simulated $V_d$ (statistical mean of vertical velocity) is in close agreement with the theoretical values from Zhang et al. (2014) (Fig. 12a), validating the reliability of the Lagrangian particle tracking method for snowfall particle transport simulations. Fig. 12b shows that the theoretical $V_d$ of snowfall particles increases with increasing particle diameter, but the dimensionless settling velocity $V_d/u_*$ decreases as $u_*$ increases, indicating that enhanced turbulence suppresses the settling efficiency of snowfall particles. The particle settling time is further calculated as $T_d = h/V_d$, where $h$ is the release height of the snowfall particles. As

shown in Figure 13, the dimensionless $T_d$ for snowfall particles in this diameter range exhibits negative exponential decay with increasing $D_p$, indicating that larger-diameter snowfall particles, due to their higher $V_d$, can rapidly complete vertical settling, whereas smaller-diameter snowfall particles have lower $V_d$ and are more susceptible to turbulent disturbances, resulting in slower settling and spatiotemporal instability. Meanwhile, an increase in $u_*$ enhances the particle retention effect, making particles more likely to remain trapped for extended periods in low-kinetic-energy regions or eddy structures, significantly

impacting the settling and transport processes.

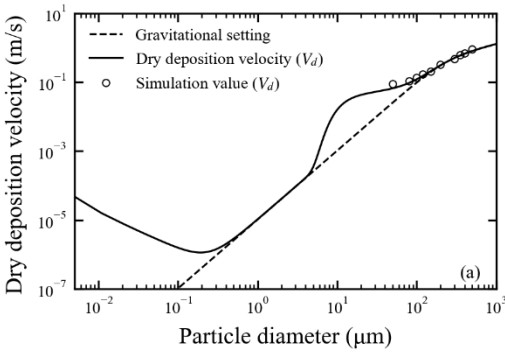 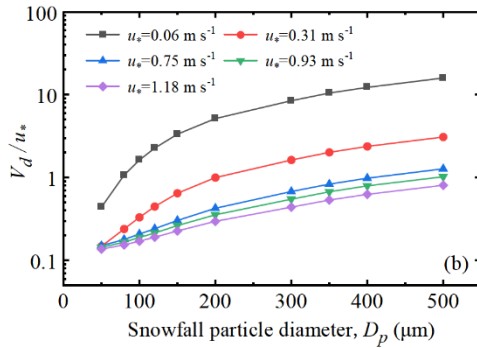

**Figure 12. (a) Comparison of simulated and theoretical snowfall particle deposition velocities at $u_*=0.75$m/s, and (b) $V_d/u_*$ as a function of particle diameter (Zhang et al. (2014)).**

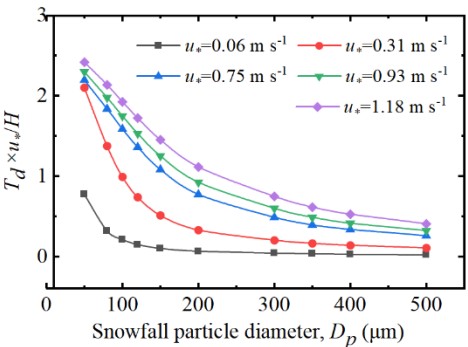

**Figure 13. Dimensionless settling time of snowfall particles as a function of $D_p$ for different $u_*$.**



To quantify the average motion distance of snowfall particles relative to the wind field during their settling time, namely the relative motion distance $Sr$:

$$Sr = Vr \cdot T_d = Vr = V_t (0.97 + 19.85 e^{-57.6 D_p}) e^{2.27 e^{-11.02 D_p} \cdot u_*} (\frac{h}{V_d}),$$  (22)

the calculations are based on the release height $h$=100 m and the definition of relative travel distance presented in this study.

The results indicate that the vertical relative travel distance ($Sr_1$) of snowfall particles decreases monotonically with decreasing particle diameter. Further analysis incorporating the influence of friction velocity reveals that for $D_p > 200$ μm, $Sr_1$ increases with increasing friction velocity, suggesting that larger-diameter particles exhibit a greater vertical motion range in stronger airflow. In contrast, for $D_p \leq 200$ μm, $Sr_1$ decreases with increasing friction velocity, indicating that smaller-diameter snowfall particles are easily entrained by turbulent eddies, displaying strong flow-following behaviour. Their lower relative velocity further affects $Sr_1$, revealing the unique motion characteristics of smaller-diameter snowfall particles in the turbulent boundary

layer. The horizontal relative travel distance ($Sr_2$) decreases with increasing $D_p$ (Fig. 14b): smaller particles are more readily transported over long distances by the airflow, resulting in larger $Sr_2$, which increases significantly with $u_*$. Further analysis of Fig. 15a shows that the ratio $Sr_1/Sr_2$ increases monotonically with the parameter $\alpha_d$. The critical state ($Sr_1/Sr_2 \approx 1$) corresponds to an $\alpha_d$ value of 0.2, indicating that under turbulent conditions, snowfall particles with $\alpha_d$ =0.2 achieve a maximum stable

motion state under the combined effects of turbulence and gravitational settling—neither settling too rapidly due to excessive gravity nor remaining suspended too long due to strong turbulent entrainment. If $\alpha_d > 0.2$, vertical relative motion dominates, with the dominance becoming more pronounced as the parameter increases; if $\alpha_d < 0.2$, horizontal relative motion predominates. Figure 15b demonstrates that the total relative travel distance ($Sr$) of snowfall particles decreases in a negative exponential manner with increasing $D_p$: the motion of larger-diameter particles is dominated by gravity, with $Sr$ approaching $h$ and showing little sensitivity to changes in $u_*$; for smaller-diameter particles, $Sr$ increases significantly with $u_*$due to enhanced turbulence,

which extends their retention time and motion path in the wind field.

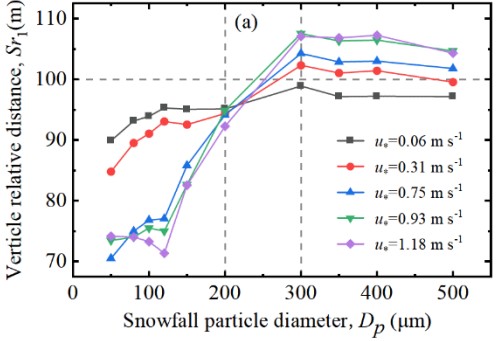
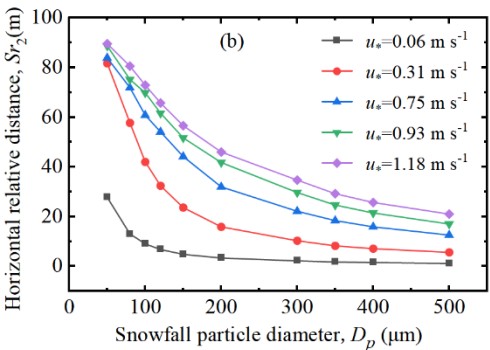

**Figure 14. Snow particle (a) $Sr_1$, and (b) $Sr_2$ as a function of $D_p$ for different $u_*$.**





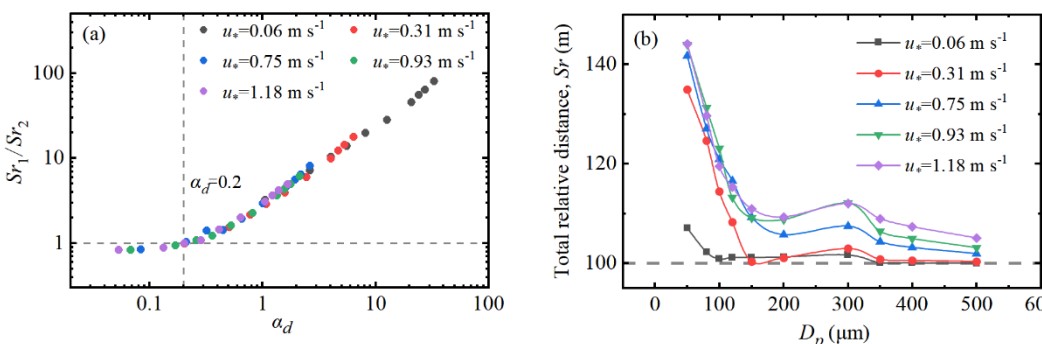

**Figure 15. (a) Ratio of $Sr_1$ to $Sr_2$ for snowfall particle as a function of the particle parameter $\alpha_d$, and (b) $Sr$ of snowfall particle as a function of $D_p$ for different $u_*$.**

### 3.3 Analysis of the impact of snowfall particle dynamic characteristics on dust wet deposition

Based on Slinn's theory (Shao, 2008; Slinn, 1984), raindrops and dust particles exhibit relative motion in the atmosphere due to differences in their terminal velocities. Suppose a raindrop falls with a terminal velocity $V_t$ and a dust particle falls with

a terminal velocity $v_t$. Then, the relative speed at which the raindrop approaches the dust particle is $(V_t - v_t)$. If the number density (number of particles per unit volume) of dust particle is $n$, then during the time interval $\Delta t$ ,the total number of dust particles that can be captured by the raindrop is $n(V_t - v_t)(\Delta t)\pi(R + r)^2$ . The number of particles actually captured by the raindrop is:

$$q = e_s n\pi(R+r)^2(V_t - v_t)\Delta t, \tag{23}$$

where the collection efficiency $e_s$ is a function of both dust radius($r$) and raindrop radius($R$).

Existing studies commonly adopt the equivalent collision efficiency assumption (Pruppacher et al., 1998), implying that when a snowfall particle comes into contact with dust, the collision directly results in the capture of the dust. According to the wet deposition formula (Eq. 23), the total number of particles collected by a snowfall particle as it settles from the air to the ground, the collection amount $Q$ can be expressed as:

$$Q = e_s n\pi(R+r)^2\sum_{i=0}^{T_d}(V_t - v_t)\Delta t_i, \tag{24}$$

where $\sum_{i=0}^{T_d}(V_D - v_d)\Delta t_i$ represents the relative motion distance of snowfall particle relative to dust particles. We assume that Aitken mode dusts ($d_p < 20$ nm) with a concentration of $n=1.5\times10^4$ cm$^{-3}$ is uniformly distributed within the boundary layer and fully follows the flow field motion, such that the velocity of dust particles equals the wind field velocity at their location, i.e., $v_t = u_{f@p}$. Therefore, the $Sr$ is also defined as the distance of relative motion between snowfall particles and dust particles.

The collection amount is the number of dust particles collected within the volume swept by the snowfall particles during their falling process, as illustrated schematically in Figure 16a. Consequently, Eq. (24) can also be expressed as:




$$Q = e_s n\pi(R+r)^2 Sr = e_s n\pi(R+r)^2 Vr \cdot T_d, \tag{25}$$

we only discuss the ultimate impact of the relative motion between snowfall particles and air during their falling on the dust

collection amount. Therefore, it is assumed that snow grains can fully collect all dust particles along their trajectories, i.e., the

collection efficiency $e_s=1$.

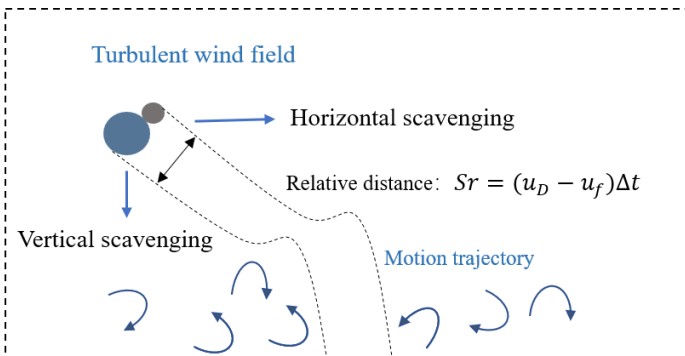

**Figure 16. Schematic diagram of the dust collection process by a falling snowfall particle.**

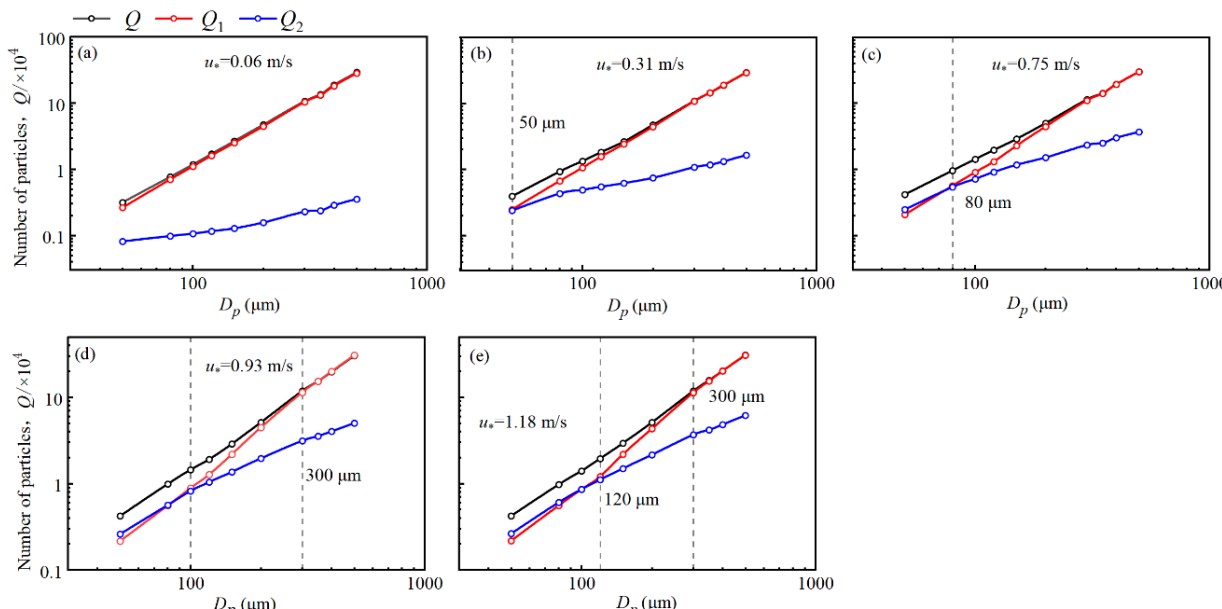

**Figure 17. Collection amount of dust by snowfall particles in various directions as a function of $D_p$ for different $u_*$.**

The number of dust particles collected by a single snowfall particle during its falling is calculated using the wet deposition

formula (Eq. 25). Fig. 17 reveals the influence of snowfall particle diameter on the total collection amount ($Q$) and its

directional components under different friction velocities : when $u_* \leqslant 0.06$ m/s, the motion of all $D_p$ snowfall particles is

dominated by vertical settling, resulting in $Q$ being almost entirely contributed by the vertical collection amount ($Q_1$). As $u_*$





increases to 1.18 m s$^{-1}$, large-diameter snowfall particles still exhibit vertical collection dominance, with $Q_1$ approaching $Q$ for

these particles; whereas small-diameter snowfall particles ($D_p$<120 μm) shift to horizontal collection dominance ($Q_2$). Moreover, the critical diameter (vertical dashed line) increases from 50 μm to 120 μm with increasing friction velocities. These results demonstrate that increasing $u_*$ drives the snowfall particle collection mechanism from vertical dominance to horizontal dominance: large-diameter snowfall particles retain vertical dominance due to gravitational settling, while small-diameter snowfall particles exhibit enhanced horizontal collection capability owing to intensified turbulent diffusion and

prolonged settling time. As shown in Figure 18, snowfall particles with $\alpha_d \geq 1$ have a $Q_1$ proportion exceeding 75%, whereas for snowfall particles with $\alpha_d$ <0.2, the horizontal collection capability significantly strengthens with increasing $u_*$ (with the minimum proportion of $Q_2$ in total collection being approximately 50%). Therefore, enhanced $u_*$ markedly boosts the horizontal collection capability of small-diameter snowfall particles.

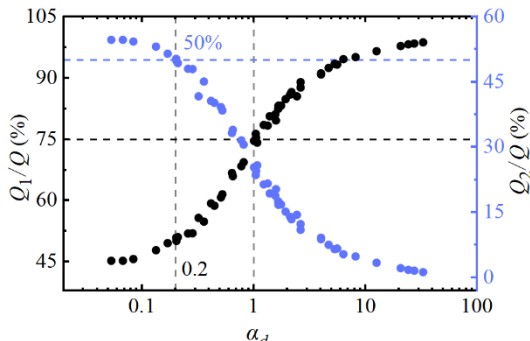

**Figure 18. Ratio of $Q_1$ and $Q_2$ in snowfall particles to the total collection amount as functions of $\alpha_d$ for different friction velocities.**

Compared to gravitational settling, the instantaneous velocity fluctuations in turbulent wind fields significantly increase the complexity of interactions between snowfalls and the flow field. Turbulence causes snowfall particles to experience regions of varying velocities, resulting in acceleration, deceleration, or entrainment into vortices, thereby affecting their motion trajectories and dust collection capability. We define the snowfall particles collection amount growth rate as the collection

enhancement efficiency relative to the gravitational settling, to quantify the impact of turbulence on the snowfall particles collection capability. The snowfall particle collection amount growth rate $r$:

$$r = \frac{Q_{u_*} - Q_0}{Q_0}, \qquad (26)$$

where $Q_{u*}$ and $Q_0$ represent the dust collection amounts by snowfalls under friction velocity and gravitational settling conditions, respectively. The results indicate that the snowfall particle collection amount growth rate exhibits a decreasing

trend with increasing snowfall particle parameter $\alpha_d$. When $\alpha_d \geq 4.5$, $r \approx 0$; whereas when $\alpha_d \leq 0.5$, the $r$ can reach approximately 50% (Fig. 19), indicating that the collection behaviour of large-diameter snow grains is primarily dominated by gravity, with

limited influence from turbulence; under higher $u_*$, snowfall particles with $\alpha_d \leq 0.5$ experience prolonged settling time due to turbulent effects, resulting in a significant enhancement of collection capability. Meanwhile, the snowfall particle collection amount growth rate increases with $u_*$. Further fitting analysis shows that $r$ decreases as a negative exponential function with increasing dimensionless parameter $\alpha_d$, which can be expressed as:

$$r = 0.26 \cdot \left( \alpha_d \right)^{-1.20}. \tag{27}$$

The coefficient of determination ($R^2$) for the collection amount growth rate exceeds 0.99, thereby establishing a mathematical model for the snowfall collection amount growth rate for dust. Given $u_*$ and $D_p$, it can effectively predict the enhancement in snowfall collection capacity attributable to turbulent effects relative to gravitational settling.

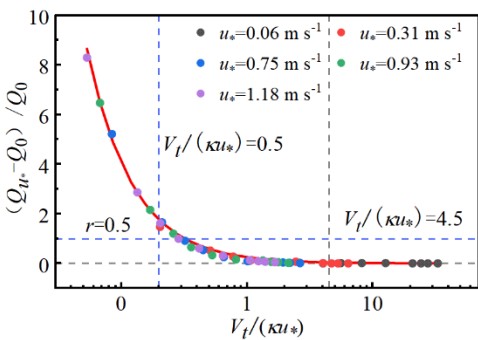

**Figure 19. Growth rate of collection amount by snowfall particle as a function of the parameter $\alpha_d$ (symbols represent simulated values, and solid lines represent fitted values).**

Snowfall particles settle in the form of a specific particles number size distribution during snowfall, which directly influences their scavenging effect on atmospheric dust. The snowfall particles size spectra is affected by ambient temperature, particle habit, precipitation intensity, and the stage of cloud and precipitation development (Harimaya et al., 2004; Woods et al., 2008). In practical applications, empirical formulas derived from raindrop size spectra are commonly used to approximate the size distribution of natural snow (Woods et al. 2008), among which the exponential distribution is widely applied in cloud microphysics to describe the snowfall particles size spectrum (Feng 2009; Solomon et al. 2009), with its basic form expressed as:

$$N(D_p) = N_{0e} \exp(-\beta_e D_p), \tag{28}$$

where $N_{0e}$ is the intercept parameter, representing the theoretical concentration when snowfall particle diameter approaches zero; $\beta_e$ is the slope parameter, controlling the decay rate of the size distribution. Referring to the classification of daily precipitation intensity during the snow season by Suriano et al. (2023) ('Light': ≤50.8 mm, 'Moderate': 50.8 < $R$ ≤ 152.4 mm, 'Heavy': 152.4 < $R$ ≤ 254.0 mm, 'Extreme': >254.0 mm), we set different precipitation intensities $R$ (2, 6, 8.5, 15, 20 mm h$^{-1}$) to systematically analyse the wet deposition efficiency of snowfall particle populations on dust. As shown in Fig. 20, for the





same particle size, the snowfall particle number concentration increases with increasing $R$, with the total number concentration spanning two orders of magnitude (Zhang et al., 2013), and exhibits exponential decay with increasing $D_p$. The variation of snowfall particle number concentration with $D_p$ and $R$ can be expressed as:

$$N_1 = (-0.8 + 42R) \times 0.986^{D_p} \times 10^6, \tag{29}$$


$$Q_{N_1(D_p)} = e_s n\pi(R+r)^2 (Vr \cdot T_d) N_1. \tag{30}$$

Based on the previously established results for single snowfall particle collection amounts, the total collection amount of the snowfall particle populations can be calculated using Eq. (30). As shown in Fig. 21a, for snowfall particles with diameters less than 200 μm, the collection amount is highly sensitive to $u_*$ and primarily dominated by turbulence: under high $u_*$ conditions, turbulence prolongs the residence time of small-diameter snowfall particle populations in the atmosphere, thereby enhancing

dust collision efficiency and significantly increasing their collection amount. In contrast, for snowfall particle populations larger than 200 μm, the collection amount is mainly dominated by inertia, with weaker influence from turbulence; the collection amount curves under different $u_*$ tend to converge, indicating reduced dependence of large-diameter snowfall particle on turbulence. Within the $u_*$=0.06-1.18 m s$^{-1}$ range, the collection amount of snowfall particles in the 100-150 μm diameter interval reaches its peak, suggesting that this size range achieves optimal scavenging efficiency under the given precipitation intensity.

Fig. 21b shows that $Q$ is linearly positively correlated with $R$, indicating that increasing precipitation intensity can significantly enhance the dust scavenging capability of snowfall particle populations. Among snowfalls of different $D_p$, the populations in the 100-150 μm range exhibits the largest slope, demonstrating that its scavenging efficiency is most sensitive to changes in precipitation intensity.

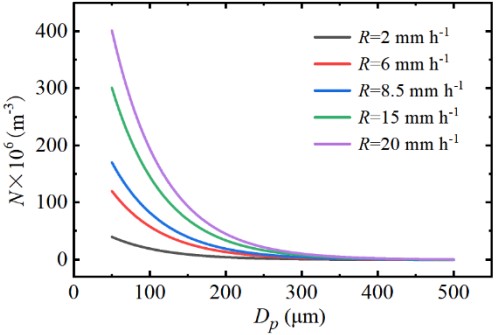

**Figure 20. Number concentration of snowfall particle as a function of $D_p$ for different precipitation intensity.**





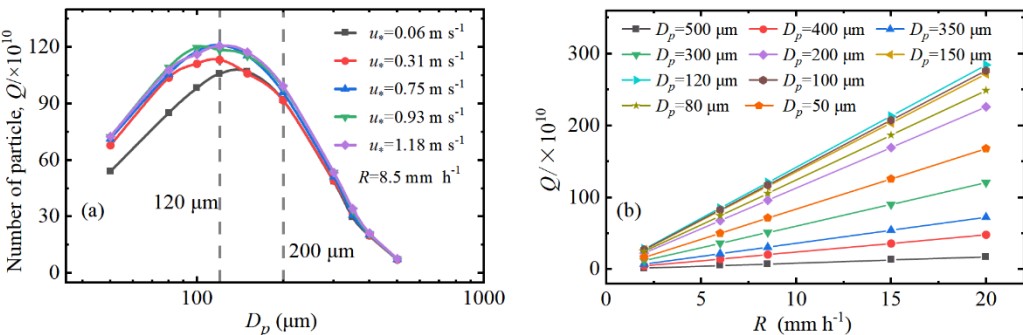

**Figure 21. Collection amount by snow particles population as a function of (a) $D_p$ for different $u_*$ at $R$=8.5 mm h$^{-1}$, and (b) $R$ for snowfall particles at $u_*$=0.75 m s$^{-1}$, respectively.**

## 3. Conclusion

In this study, we investigated the motion of snowfall particles in a turbulent boundary layer using the Eulerian-Lagrangian approach. The mechanism for dust collection by snowfall is investigated by analysing the snowfall particles behaviours and its dependence on turbulent characteristics. The main conclusions are as follows:

This study reveals the combined influence of particle diameter and friction velocity ($u_*$) on snow particle dynamics characteristics. The findings demonstrate that turbulence significantly prolongs the residence time of small snowfall particles in the atmosphere, thereby enhancing the randomness of their trajectories and consequently increasing the uncertainty in their final deposition. Furthermore, turbulence affects the particles' relative motion distance to the air: large particles, governed primarily by gravity, maintain stable vertical trajectories with the total relative motion distance close to the release height. In contrast, small particles experience a significantly expanded horizontal range due to turbulence, leading to a substantial increase in their relative motion distance. Overall, the total relative motion distance of snowfall particles exhibits a negative exponential decay with increasing particle diameter. As friction velocity increases, the dominance of vertical and horizontal motion of snowfall particles shifts: The ratio of the vertical relative motion distance ($Sr_1$) to the horizontal relative motion distance ($Sr_2$) of snowfall particles increases monotonically with the parameter $\alpha_d$, where the critical state ($Sr_1/Sr_2 \approx 1$) corresponds to a snow particle $\alpha_d$ value of 0.2, indicating that snow particles with $\alpha_d = 0.2$ can maintain long-term motion stability under turbulent conditions. If this parameter exceeds 0.2, vertical relative motion dominates, with the dominance becoming more pronounced as the parameter increases; if $\alpha_d \leq 0.2$, the horizontal relative motion exerts the primary influence. The influence mechanism of snowfall particle dynamic characteristics in turbulent environments on dust wet deposition indicates that the scavenging efficiency of snowfall particles for dust is significantly different from gravitational settling: when the snow particle parameter $\alpha_d \geq 4.5$, the collection growth rate($r$) approaches zero, and the collection behaviour is entirely dominated by gravity; whereas when $\alpha d \leq 0.5$, the $r$ can reach approximately 50%, with turbulence significantly enhancing the collection capacity of small snow particles. With increasing friction velocity, the dust collection capability of snowfall



particles is markedly enhanced, and the collection mechanism shifts from predominantly vertical to mainly horizontal. Snowfall particles with $\alpha_d \geq 1$ maintain a vertical collection advantage due to gravitational settling (accounting for over 75% of the total collection); while those with $\alpha_d \leq 0.2$ shift to horizontal collection dominance under turbulence effects (accounting for at least 50%). Within the $u_*$ range of 0.06–1.18 m s$^{-1}$, the scavenging efficiency of 100–150 μm snowfall particle ensembles is optimal and exhibits a linear positive correlation with precipitation intensity. Among these, 100–150 μm snowfall particles are the most sensitive to variations in precipitation intensity.

A predictive model for snowfall particle motion (Eq. 20) is established, providing a quantitative theoretical basis for snowfall particle behaviour in wind-snow two-phase flow. A quantitative formula for wet deposition (Eq. 25) is also proposed, which can be utilized to quantify the enhancement effect of turbulence on the collection capability of snow particles. Consequently, the atmospheric dust wet deposition flux is accurately predicted, and significant application value is demonstrated in the fields of artificial dust removal and environmental assessment.

**Acknowledgments**

This work was supported by the National Natural Science Foundation of China (grant No. 42376232), the NSFC-FDCT Joint Research Program (Grant No. 42381164666), National Natural Science Foundation of China (No. 52306197). Computing resources are supported by the Supercomputing Center of Lanzhou University.

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
