# Peer review of "Dynamic characteristics of snowfall particles in atmospheric turbulent boundary layer and its effect on dust wet deposition"

_EGUsphere, 2025_

## Referee Comment (RC1)

**Review of "Dynamic characteristics of snowfall particles in atmospheric turbulent boundary layer and its effect on dust wet deposition" by Zhang et al.**

This study employs a Delayed Detached-Eddy Simulation (DDES) coupled with a Lagrangian particle tracking method to investigate the motion of snowfall particles within an atmospheric turbulent boundary layer. The authors analyze how turbulence influences the relative motion between snow particles and air, identifying a critical dimensionless parameter ($\alpha_d = 0.2$). The results suggest that for ($\alpha_d < 0.2$), horizontal relative motion dominates, significantly enhancing the swept volume and, consequently, the potential for dust wet deposition compared to gravitational settling alone.

The manuscript addresses an interesting and complex problem in atmospheric physics, the interplay between turbulence and precipitation scavenging. The use of DDES to resolve the turbulent wind field represents a sophisticated approach compared to standard RANS models often used in this field. The identification of the transition threshold at $\alpha_d = 0.2$ offers a potentially valuable metric for parameterizing wet deposition in larger-scale models.

Please note that my evaluation focuses on the physical interpretation of the snowfall dynamic characteristics, the experimental design regarding particle physics, and the implications for dust wet deposition. As I am not a specialist in Computational Fluid Dynamics or hybrid RANS/LES modeling, I have not critically assessed the numerical implementation of the DDES model, the specific grid convergence strategies, or the stability of the solution. My comments regarding the methodology are restricted to its physical justification and consistency with atmospheric principles.

While the trajectory analysis appears rigorous, I have concerns regarding the physical simplifications made for the snow particles, specifically the assumptions regarding particle shape and collection efficiency, which likely lead to an overestimation of the deposition flux. These issues should be addressed before publication.

**Major comments**

1. In Section 3.3, the calculation of the dust collection amount assumes that "snow grains can fully collect all dust particles along their trajectories, i.e., the collection efficiency $e_s = 1$". This is a very strong assumption that likely leads to an overestimation of the removal rate. Collection efficiency can be governed by aerodynamic effects (Brownian diffusion, interception, and inertial impaction). For the Aitken mode dust mentioned in the text, flow streamlines around the falling snow particle may carry aerosols away from the collector surface, resulting in efficiencies well below 1.0. Could the authors explicitly discuss the magnitude of uncertainty introduced by this assumption. The results should

perhaps be framed as "maximum potential encounter volume" rather than actual "collection amount."

2. The study simplifies snowfall particles as spheres with a density of 340 kg/m3. While this simplifies the Lagrangian tracking, natural snow particles (dendrites, plates, aggregates) exhibit different drag coefficients and terminal velocities compared to spheres. This aerodynamic difference directly affects the calculation of Vt and the critical parameter $\alpha_d$. A discussion is needed on how non-spherical drag would alter the $\alpha_d$ threshold. Would complex shapes be more or less susceptible to the horizontal entrainment described in this study?

3. The manuscript defines the relative motion distance Sr as the product of the relative velocity and the suspension time Td. While this metric is useful for comparing cases, it is essentially a proxy for the "swept volume" or "effective path length." The text essentially equates longer suspension time in turbulence with higher deposition. However, if a particle is trapped in a vortex, it may be "sweeping" the same volume of air repeatedly (which has already been scavenged), rather than encountering fresh dust. Please clarify if the model accounts for the depletion of dust in the local trajectory of the snow particle, or if it assumes the dust concentration remains constant regardless of how many times the snow particle passes through a specific eddy.

**Specific comments:**

1. Some figure captions can be enhanced by defining the variables/parameters in the figure. For instance, Figure 17, the readers have to read through the manuscript to understand the meaning of Q, Q1, and Q3. And the meaning of the lines in Figure 8.

2. Line 71: the statement here should be softened, as the two research categories described do not fully encompass all possible approaches. Additional types of studies may exist.

3. Line 148: Add a comma before "and."

4. Figure 6: What do in panel (b) mean?

5. Equations 18 and 19. Please elaborate on why this particular functional form was chosen. A brief justification or reference would help readers better understand the reasoning behind these expressions.

6. Equation 20, Please check whether a bar is missing over the variable on the right-hand side of the equation?

7. In Figure 14(a) and the y-axis label, the word "Verticle" is misspelled. It should be corrected to "Vertical."

---

## Author Comment (AC1)

**Response to Reviewer**

*This study employs a Delayed Detached-Eddy Simulation (DDES) coupled with a Lagrangian particle tracking method to investigate the motion of snowfall particles within an atmospheric turbulent boundary layer. The authors analyze how turbulence influences the relative motion between snow particles and air, identifying a critical dimensionless parameter ($\alpha_d = 0.2$). The results suggest that for ($\alpha_d < 0.2$), horizontal relative motion dominates, significantly enhancing the swept volume and, consequently, the potential for dust wet deposition compared to gravitational settling alone.*

*The manuscript addresses an interesting and complex problem in atmospheric physics, the interplay between turbulence and precipitation scavenging. The use of DDES to resolve the turbulent wind field represents a sophisticated approach compared to standard RANS models often used in this field. The identification of the transition threshold at $\alpha_d = 0.2$ offers a potentially valuable metric for parameterizing wet deposition in larger-scale models.*

*Please note that my evaluation focuses on the physical interpretation of the snowfall dynamic characteristics, the experimental design regarding particle physics, and the implications for dust wet deposition. As I am not a specialist in Computational Fluid Dynamics or hybrid RANS/LES modeling, I have not critically assessed the numerical implementation of the DDES model, the specific grid convergence strategies, or the stability of the solution. My comments regarding the methodology are restricted to its physical justification and consistency with atmospheric principles. While the trajectory analysis appears rigorous, I have concerns regarding the physical simplifications made for the snow particles, specifically the assumptions regarding particle shape and collection efficiency, which likely lead to an overestimation of the deposition flux. These issues should be addressed before publication.*

**Response:** We sincerely thank the reviewer for taking the time to review our manuscript and for providing valuable and insightful comments. In response to the detailed and profound suggestions, we have comprehensively revised and supplemented the manuscript to enhance the rigor of the research and the clarity of its presentation. Here, we provide detailed responses to each of the questions and

suggestions raised by the reviewer. All comments and suggestions have been carefully considered. Below, the reviewer's original comments are presented in *italics*, and our responses are shown in blue.

**In response to the reviewers' concerns about the physical assumptions in this paper, we provide the following detailed explanations:** this study focuses on theoretical analysis, with the core research content being the effects of turbulent characteristics of the atmospheric boundary layer on the motion of snow particles in the air, and further explores the scavenging effect of dust particles caused by the relative motion between snow particles and air. To highlight the research focus on the impact of turbulence on snow particle motion and facilitate targeted analysis, we have made reasonable physical simplifications for the shape and collection efficiency of snow particles, which do not affect the scientificity and rationality of the core research conclusions.

The research results of this paper confirm that the turbulent characteristics of the atmospheric boundary layer have an important impact on the snowfall wet deposition process. This conclusion provides theoretical support for in-depth interpretation of the physical mechanism of snowfall wet deposition and for conducting more accurate wet deposition assessments in the future, which is of great academic significance.

It should be noted that the quantitative results of wet deposition amount obtained in this paper cannot be directly applied to the quantitative analysis of snowfall wet deposition in actual environments for the time being. To improve the universality of the research conclusions in this paper, we have specially introduced the dimensionless parameter $\alpha_d$, which can comprehensively characterize the particle dynamic characteristics and the turbulent characteristics of the boundary layer. In the subsequent research plan, we will focus on exploring the intrinsic relationship between snow particle shape and the $\alpha_d$ parameter, and carry out special research on the collection efficiency of snow particles with different shapes and size, so as to gradually improve the research content, promote the better application of the research results of this paper in practical environments, and make up for the limitations of the current research.

In the revised version, we will provide a more explicit explanation of the premise assumptions of this work and the scope of application of the conclusions to avoid misunderstandings among readers.

**Major comments**

1. *In Section 3.3, the calculation of the dust collection amount assumes that "snow grains can fully collect all dust particles along their trajectories, i.e., the collection efficiency e = 1". This is a very strong assumption that likely leads to an overestimation of the removal rate. Collection efficiency can be governed by aerodynamic effects (Brownian diffusion, interception, and inertial impaction). For the Aitken mode dust mentioned in the text, flow streamlines around the falling snow particle may carry aerosols away from the collector surface, resulting in efficiency well below 1.0. Could the authors explicitly discuss the magnitude of uncertainty introduced by this assumption. The results should perhaps be framed as "maximum potential encounter volume" rather than actual "collection amount."*

   **Response:** Thank reviewer for the suggestion. We fully agree that the assumption of setting the collection efficiency to 1 may overestimate the actual scavenging efficiency. This simplification was indeed adopted to focus on the impact mechanism of turbulence on the relative motion between snow particles and dust particles. Following your core recommendation, we have revised the term "actual collection amount" throughout the manuscript to "maximum potential scavenging volume". In the relevant sections, we have also added clarifications regarding the limitations of this idealization, specifically noting that the actual collection efficiency may fall below the theoretical value due to aerodynamic effects. All corresponding revisions have been incorporated in the revised version.

2. *The study simplifies snowfall particles as spheres with a density of 340 kg/m3. While this simplifies the Lagrangian tracking, natural snow particles (dendrites, plates, aggregates) exhibit different drag coefficients and terminal velocities compared to spheres. This aerodynamic different directly*

*affects the calculation of Vt and the critical parameter $\alpha_d$. A discussion is needed on how non-spherical drag would alter the $\alpha_d$ threshold. Would complex shapes be more or less susceptible to the horizontal entrainment described in this study?*

**Response:** Thank reviewer for the suggestion. The dimensionless parameter $\alpha_d$ we employed is essentially defined to characterize particle dynamics through the terminal settling velocity, which inherently integrates the influence of physical properties such as particle shape and density on drag. Thus, even when considering non-spherical particles, their aerodynamic effects are already incorporated into the definition of this parameter through the core variable of terminal settling velocity. If changes in shape alter the settling velocity, the value of $\alpha_d$ will adjust accordingly, thereby naturally accounting for the influence of particle shape within the existing framework.

The specific effects of complex shapes on the α threshold and horizontal scavenging sensitivity are indeed important questions that warrant in-depth investigation. However, this involves specialized research on the coupling between microscopic particle morphology and flow fields, which extends beyond the core scope of this paper focusing on the macroscopic mechanisms of turbulence. Currently, our team is conducting dedicated studies on the α values corresponding to different snow particle shapes, and the relevant findings will be reported in subsequent work.

3. *The manuscript defines the relative motion distance Sr as the product of the relative velocity and the suspension time Td. While this metric is useful for comparing cases, it is essentially a proxy for the "swept volume" or "effective path length." The text essentially equates longer suspension time in turbulence with higher deposition. However, if a particle is trapped in a vortex, it may be "sweeping" the same volume of air repeatedly (which has already been scavenged), rather than encountering fresh dust. Please clarify if the model accounts for the depletion of dust in the local trajectory of the snow particle, or if it assumes the dust concentration remains constant regardless of how many times the snow particle passes through a specific eddy.*

**Response:** Thank reviewer for the suggestion. This study is grounded in the understanding of

atmospheric boundary layer physical processes, where the probability of a single snow particle being continuously trapped within the same vortex is extremely low. The Stokes number of snow particles is typically several orders of magnitude larger than that of dust particles, and their greater inertia makes it difficult for them to be persistently captured by small-scale eddies. Therefore, it is reasonable for the model to assume that snow particles continuously encounter unswept air while moving through turbulence. Furthermore, the intense turbulent mixing within the boundary layer rapidly replenishes removed dust particles, maintaining a relatively constant local dust concentration. Consequently, based on the significant inertial disparity between snow and dust particles and the strong mixing effects in the boundary layer, we believe the scenario of "repetitive scavenging" raised by the reviewer is unlikely to occur in the physical processes examined in this study.

**Specific comments:**

1. *Some figure captions can be enhanced by defining the variables/parameters in the figure. For instance, Figure 17, the readers have to read through the manuscript to understand the meaning of Q, Q1, and Q2. And the meaning of the lines in Figure 8.*

   **Response:** Thank reviewer for the suggestion. Based on the reviewers' suggestions above, we have supplemented and revised the captions of several figures throughout the manuscript. For example, in the relevant illustrations, it is now clearly indicated that $Q$ denotes the total maximum potential scavenging volume, $Q_1$ represents the vertical maximum potential scavenging volume, and $Q_2$ refers to the horizontal maximum potential scavenging volume. Additionally, the meanings of the line styles in Figure 8 have been specified: the black solid line represents the wind speed at the snow particle location, the green solid line indicates the particle velocity, and the blue solid line corresponds to the modulus of the difference between the wind speed at the particle position and the particle velocity.

2. *Line 71: the statement here should be softened, as the two research categories described do not fully encompass all possible approaches. Additional types of studies may exist.*

**Response:** Thank reviewer for the suggestion. According to the reviewer's comments, we have softened the statement in line 71 and revised it to the following:"Currently, relevant research can primarily be divided into two type. The first type uses field-collected snow sample data to establish relationships between precipitation, snowfall particle size distribution, and ground-level aerosol mass concentration, thereby evaluating the scavenging efficiency of snowfall on aerosol. The second type focuses on the collision scavenging mechanisms between a single snowflakes (or snow crystals) falling at terminal velocity $V_D$ and aerosol particles, investigated through laboratory simulations or direct observations (Pruppacher et al. 1998). It should be noted that while these two approaches are widely represented in current literature, they do not exhaust all possible research approaches."

3. *Line 148: Add a comma before "and."*

Response: Thank reviewer for the suggestion. We have made modifications on line 148.

4. *Figure 6: What do in panel (b) mean?*

**Response:** Thank reviewer for the suggestion. Thank you for pointing out the lack of clarity in Figure 6(b). Figure 6(b) aims to illustrate the following key findings: the simulated turbulence intensity profile generally aligns with the lower bound of the range suggested by the ASCE 7-III standard (which allows turbulence intensity values to vary within ±20% of a specified baseline value [Kozmar, 2011]), with a maximum observed relative error of 16% in the near-ground region. Furthermore, the overall profile lies between the upper limits specified by the Chinese national standard and the ASCE 7-III standard. This demonstrates that the simulation results effectively reproduce the wind field characteristics of a real atmospheric boundary layer.

We have revised the relevant description in the main text of the manuscript (page 10), which now reads: "As shown in Figure 6(b), the simulated turbulence intensity profile largely aligns with the lower bound of

the range recommended by the ASCE 7-III standard (which permits variations within ±20% of the baseline value [Kozmar, 2011]), with a maximum relative error of 16% in the near-ground region. Overall, the profile lies between the upper limits specified by the Chinese national standard and the ASCE 7-III standard. Thus, the simulation results effectively reproduce the wind field characteristics of a real atmospheric boundary layer."

5. *Equations 18 and 19. Please elaborate on why this particular functional form was chosen. A brief justification or reference would help readers better understand the reasoning behind these expressions.*

**Response:** Thank reviewer for the suggestion. The model takes the friction velocity $u_*$ (the fundamental driving force for particle transport) as the core independent variable. Equation (18) first constructs the dimensionless ratio $Vr/Vt$ to eliminate the dominant influence of gravitational settling, thereby placing particles of different sizes on a unified dynamic comparison baseline. It employs an exponential function form dependent on particle diameter $Dp$ (e.g., $e^{(-kDp)}$) to characterize the nonlinear decay behavior of particle inertia with increasing particle size: for smaller particles with lower Stokes numbers, motion is dominated by turbulent disturbances, making them highly sensitive to changes in $u_*$, and the relevant parameters decay rapidly with increasing $Dp$; for larger particles with higher Stokes numbers, inertial effects become dominant, leading to a significantly reduced sensitivity to changes in $u_*$ and a much slower decay rate. This transition corresponds physically to the variation in Stokes number, while the exponential function naturally captures this nonlinear dynamic process mathematically. Thus, the model integrates $u_*$ as the dynamic input, the dimensionless ratio as a unified benchmark, and the exponential function to represent the particle inertia effect, forming a complete physical chain that spans from external driving and dynamic balance to particle response—combining clear dynamic mechanisms with concise mathematical expression.

In Equation (19), the standard deviation of the relative velocity fluctuations $Vr'$ is expressed as a linear function of the friction velocity $u_*$, which primarily originates from the scaling laws of boundary-layer turbulence and the response mechanisms of particle dynamics. In near-surface turbulence, the root mean square of velocity fluctuations is generally proportional to $u_*$, reflecting the fundamental physical

relationship that turbulent fluctuation energy derives from surface shear stress. The linear form $Vr' = c0 + k(Dp) \cdot u_*$ intuitively captures the quasi-linear response of particle fluctuations to changes in turbulent intensity.

6. *Equation 20, Please check whether a bar is missing over the variable on the righthand side of the equation?*

   Response: Thank reviewer for the suggestion. We have added a bar above Equation (20) and the relevant formula variables.

7. *In Figure 14(a) and the y-axis label, the word "Verticle" is misspelled. It should be corrected to "Vertical."*

   Response: Thank reviewer for the suggestion. We have corrected the y-axis label in Figure 14(a) from "Verticle" to "Vertical".